# High-Grade B-Cell Lymphoma (HGBL) with *MYC* and *BCL2* and/or *BCL6* Rearrangements Is Predominantly BCL6-Rearranged and BCL6-Expressing in Taiwan

**DOI:** 10.3390/cancers13071620

**Published:** 2021-03-31

**Authors:** Cheng-Chih Tsai, Yung-Cheng Su, Oluwaseun Adebayo Bamodu, Bo-Jung Chen, Wen-Chiuan Tsai, Wei-Hong Cheng, Chii-Hong Lee, Shu-Min Hsieh, Mei-Ling Liu, Chia-Lang Fang, Huan-Tze Lin, Chi-Long Chen, Chi-Tai Yeh, Wei-Hwa Lee, Ching-Liang Ho, Shiue-Wei Lai, Huey-En Tzeng, Yao-Yu Hsieh, Chia-Lun Chang, Yu-Mei Zheng, Hui-Wen Liu, Yun Yen, Jacqueline Whang-Peng, Tsu-Yi Chao

**Affiliations:** 1Division of Hematology and Oncology, Department of Internal Medicine, Taipei Medical University-Shuang Ho Hospital, New Taipei City 235, Taiwan; minehoks@gmail.com (C.-C.T.); xxxx@s.tmu.edu.tw (Y.-C.S.); 16625@s.tmu.edu.tw (O.A.B.); 13520@s.tmu.edu.tw (W.-H.C.); ctyeh@s.tmu.edu.tw (C.-T.Y.); 10573@s.tmu.edu.tw (Y.-Y.H.); 19290@s.tmu.edu.tw (H.-W.L.); 2Ph.D. Program for Cancer Molecular Biology and Drug Discovery, College of Medical Science and Technology, Taipei Medical University and Academia Sinica, Taipei City 115, Taiwan; 3Department of Medical Research and Education, Taipei Medical University-Shuang Ho Hospital, New Taipei City 235, Taiwan; 4Department of Pathology, Taipei Medical University-Shuang Ho Hospital, New Taipei City 235, Taiwan; 15006@s.tmu.edu.tw (B.-J.C.); b8301130@tmu.edu.tw (C.-H.L.); s20021@skmh.tmu.edu.tw (M.-L.L.); whlpath97616@shh.org.tw (W.-H.L.); 5Department of Pathology, Tri-Service General Hospital, National Defense Medical Center, Taipei City 114, Taiwan; doc31779@mail.ndmctsgh.edu.tw; 6Department of Anatomic Pathology, Taipei Institute of Pathology, Taipei City 103, Taiwan; 7Department of Clinical Pathology, Taipei Medical University-Shuang Ho Hospital, New Taipei City 235, Taiwan; 13464@s.tmu.edu.tw; 8Department of Pathology, Taipei Medical University Hospital, Taipei City 110, Taiwan; ccllfang@tmu.edu.tw (C.-L.F.); chencl@tmu.edu.tw (C.-L.C.); 9Division of Hematology and Oncology, Department of Medicine, Taipei Medical University Hospital, Taipei City 110, Taiwan; mount591@hotmail.com (H.-T.L.); tzhuen@tmu.edu.tw (H.-E.T.); 10Department of Pathology, School of Medicine, Taipei Medical University, Taipei City 110, Taiwan; 11Division of Hematology-Oncology, Department of Internal Medicine, Tri-Service General Hospital, National Defense Medical Center, Taipei City 114, Taiwan; 02241@ndmctsgh.edu.tw (C.-L.H.); xsurfer@office365.ndmctsgh.edu.tw (S.-W.L.); 12Graduate Institute of Clinical Medicine, College of Medicine, Taipei Medical University, Taipei City 110, Taiwan; 13Program for Cancer Molecular Biology and Drug Discovery, College of Medical Science and Technology, Taipei Medical University, Taipei City 110, Taiwan; yyen@tmu.edu.tw; 14Department of Medicine, Division of Hematology and Oncology, Taipei Medical University-Wan-Fang Hospital, Taipei City 116, Taiwan; richardch9@tmu.edu.tw (C.-L.C.); lilindr3@gmail.com (Y.-M.Z.); jqwpeng@nhri.org.tw (J.W.-P.); 15Taipei Cancer Center, Taipei Medical University, Taipei City 110, Taiwan

**Keywords:** MYC, BCL2, BCL6, HGBL, DLBCL, double-hit lymphoma, non-GCB, gene rearrangement, non-Hodgkin’s lymphoma

## Abstract

**Simple Summary:**

This study highlights the epidemiological, cytogenetic and clinical difference between patients with multiple hit diffuse large B-cell lymphoma in Taiwan and those from western countries. Unlike in the West, the majority of patients with multiple hit lymphoma in Taiwan harbor a *BCL6* rearrangement. Almost three in every five *BCL6*-rearranged double hit lymphoma cases in Taiwan are non-GCB phenotype, indicating, at least in part, that the preferential screening for double hit with *BCL6* rearrangement may be a clinically-informative modality for patients with non-GCB phenotype DLBCL in Taiwan. This also suggests the need for a different treatment approach than is obtained in the West where *BCL6* double hit lymphomas are seemingly GCB. Consistent with our present findings, mandatory screening for *BCL6*-rearrangement in suspected DLBCL cases in Taiwan may aid early diagnosis, therapy decision, and clinical outcome forecast.

**Abstract:**

This study investigated the epidemiological and clinical peculiarities of *BCL2* and *BCL6* rearrangement in patients with high grade B-cell lymphoma (HGBL) from Taiwan, compared with data from Western countries. Two hundred and eighty-two DLBCL cases from Taipei Medical University-affiliated hospitals (*n* = 179) and Tri-Service General Hospital (*n* = 103) were enrolled for this study. From the 282, 47 (16.7%) had *MYC* translocation; 24 of these harbored concurrent *BCL2* and/or *BCL6* translocation (double-hit, DH or triple-hit, TH). Twelve DH-HGBL cases had simultaneous *MYC* and *BCL6* translocations, 8 harbored *MYC* and *BCL2* rearrangement, while the remaining 4 patients exhibited TH. Together, 66.7% of DH/TH-HGBL patients were *BCL6* rearrangement positive. Among these *BCL6*-rearranged DH/TH-HGBL patients, only 6 (37.5%) overexpressed MYC and BCL6 proteins simultaneously, indicating that MYC-BCL6 co-overexpression may not be plausible surrogate biomarker for screening *BCL6*-rearranged DH-HGBL. By the end of year 5, all patients with TH-HGBL, *BCL2* DH-HGBL and all but one *BCL6* DH-HGBL cases had expired or were lost to follow-up. Progression-free survival (PFS) was longer for the non-DH/TH-HGBL group compared with the DH/TH-HGBL group. While the patients with *BCL2* DH-HGBL were lost to follow-up by day 800, their remaining TH-HGBL and *BCL6* DH-HGBL peers exhibited very poor PFS, regardless of age strata. More so, patients with *BCL6* rearrangement were 5.5-fold more likely associated with extranodal involvement compared with their BCL2-rearranged peers. Moreover, ~60.0% of the *BCL6*-rearranged DH-HGBL cases were non-GCB, suggesting that including screening for *BCL6* rearrangement in patients with the non-GCB phenotype may aid medical decision-making and therapeutic strategy. Contrary to contemporary data from western countries, 2 in every 3 patients with DH/TH-HGBL in Taiwan harbor *BCL6* rearrangement. Consistent with present findings, we recommend mandatory screening for *BCL6* rearrangement in patients with aggressive HGBL in Taiwan.

## 1. Introduction

Diffuse large B-cell lymphoma (DLBCL) is the most common subtype of non-Hodgkin’s lymphoma and it accounts for about half of all lymphomas in Taiwan [1]. DLBCL is composed of a heterogeneous group of morphologically similar lymphomas with a broad prognostic spectrum. Clinical prognostication is mostly aided by gene expression profiling (GEP). Based on GEP, DLBCL can be divided into 3 biologically distinct subtypes based on cell-of-origin (COO), namely germinal center B-cell (GCB), activated B-cell (ABC) and type III [2].

The last two decades has been characterized by piqued interest in identifying and demystifying the unique subtypes of DLBCL COO, alongside the molecular features that may facilitate their use, independent of the International Prognostic Index (IPI), for patient stratification, high-risk disease group identification, and prediction of treatment failure and/or prognosis [3]. Limitations in the clinical application/adoption of the GEP-based COO identification, due to its high cost and mandatory requirement of fresh frozen tissue, has led to the use of immunohistochemisty (IHC)-based methods like the Tally and Hans algorithms for COO identification in clinical practice [3]. Howbeit, compared with GEP (the gold standard for identification of COO), the sensitivity of IHC is about 70% and 90% for the GCB and so-called “non-GCB” groups, respectively [3,4]. More recently, there has been development of new platforms, like the RNA-based Lymph2Cx assay, which are yet to be adapted for clinical use but exhibit relatively higher concordance with GEP than the IHC, are reproducible across laboratories, and are adaptable for digital GEP of fixed paraffin-embedded tissue [4,5,6]. Furthermore, apart from the COO classification of DLBCL, another RNA-based method called the comprehensive consensus clustering (CCC), which identifies prevalent B-cell receptor signaling-associated metabolic pathways and the crucial peculiarities in tumor immune/inflammatory infiltrate, has been touted for identification of distinct subtypes of DLBCL [7]. However, despite the ability of the CCC to identify important intra-DLBCL heterogeneity, its clinical application remains limited.

Moreover, in an effort to provide “a potential nosology for precision-medicine strategies in DLBCL”, Schmitz et al. reported the discovery of four principal “genetic subtypes of DLBCL with distinct genotypic, epigenetic, and clinical characteristics”, namely, MCD (characterized by co-occurrence of ***M****YD88*^L265P^ and ***CD****79B* mutations), and N1 (based on ***N****OTCH**1*** mutations) which are ABC-dominated, the EZB (rich in ***EZ****H2* mutations and ***B****CL2* translocations) which is mostly GCB, and the BN2 (harboring ***B****CL6* fusions and ***N****OTCH**2*** mutations) consisting of ABC, GCB and unclassified cases [8]. In spite of the advances in biomolecular techniques that have significantly helped to expand our understanding of the pathobiology of DLBCL, facilitating the identification of disease subsets with similar targetable bio-traits, the GEP remains the gold standard for identification of COO. While the GCB DLBCL is characterized by good prognosis with a 5-year overall survival (OS) of ~59%, the ABC DLBCL is more aggressive and associated with very poor prognosis when treated with standard chemotherapy, with a 5-year OS of ~31% [9,10].

There is also accrued evidence that DLBCL with concurrent rearrangement of *MYC* and *BCL2* or *BCL6*, (also called double-hit lymphoma (DHL)) are characterized by dismal prognosis, as evidenced by median OS < 1.5 years, regardless of COO [11]. Until date, conventional knowledge indicates that majority (90–95%) of DHL are GCB phenotype, and that while concurrent *BCL2*/*MYC* rearrangement accounts for ≥75% of DHL, the less likely *BCL6*-rearranged DHL accounts for <25% [11,12].

It has also been reported that most *BCL2*-rearranged DHL have concurrent overexpression of MYC and BCL2 detected by immunohistochemical (IHC) staining [13]. However, while the clinicopathological characteristics and probable therapeutic options of *BCL2*-rearranged DHL are well studied and documented; only few studies have focused on *BCL6*-rearranged DHL and its clinical features. In one of those few studies, Li et al. in their clinicopathological analysis of DHL, found 13 *BCL6*-rearranged and 83 *BCL2*-rearranged DHL cases from a DLBCL cohort of ~1000 patients, however they reported no dissimilarity in their clinicopathological characteristics, except that the *BCL6*-rearranged DHL was less likely to be phenotypically GCB [14]. The dearth of documented consensual clinicopathological characterization of Taiwanese patients with DHL/THL, otherwise designated high-grade B-cell lymphoma (HGBL) with MYC and BCL2 and/or BCL6 rearrangements [14,15], informs the present study. Thus, the present study (i) investigated the epidemiological peculiarity of patients with DHL in Taiwan, (ii) comparatively analyzed the cytogenetic and clinical traits of DHL cases in Taiwan and reported cohort traits from western countries, and (iii) delineates the clinicopathological characteristics with associated prognostic peculiarity of patients with DHL in Taiwan. The present study incorporates patients with DLBCL from Shuang Ho Hospital, Taipei Medical University Hospital and Wan-Fan Hospital as cohort 1, and Tri-Service General Hospital as cohort 2. Interestingly, contrary to contemporary data mostly originating from western countries, in Taiwan, the majority (68.2%) of patients with DH/TH-DLBCL harbor *BCL6* rearrangement, and MYC/BCL6 co-expression is not a surrogate marker of MYC/BCL6-rearranged DH/TH-DLBCL.

## 2. Methods

### 2.1. Sample Selection

A total of 282 patients with DLBCL diagnosed between January, 2009 and December, 2019 and with plausible cytogenetic and corresponding clinicopathological data were selected from Shuang Ho Hospital, Taipei Medical University Hospital, Wan-Fang Hospital, and Tri-Service General Hospital. In our study consortium, almost all new cases of DLBCL were tested for *MYC* rearrangements by fluorescence in-situ hybridization (FISH) analysis, and *MYC*-rearranged cases were further probed for *BCL2* or *BCL6* rearrangements. Initial classification of cases was based on the 2008 World Health Organization (WHO) classification of hematopoietic and lymphoid tissues [16], and re-classified consistent with the 2016 WHO classification guideline [17] by 2 experienced hematopathologists. No tumor specimen was positive for Epstein-Barr virus (EBV) infection. DLBCL with MYC and BCL2 and/or BCL6 rearrangements per the classic definition were defined as DHL/THL, and those with co-expression of MYC and BCL2 and/or BCL6 protein were defined as double or triple expressor lymphoma (DEL/TEL). Patients’ clinicopathological information was retrieved from digital medical records of the hematopathological departments of participating medical centers; The attending physicians determined therapy regimen for the patients, and this included R-CHOP (rituximab, cyclophosphamide, doxorubicin, vincristine, and prednisone) with or without etoposide, or R-EPOCH (rituximab, etoposide, prednisone, vincristine, cyclophosphamide, and doxorubicin).

### 2.2. Fluorescence In Situ Hybridization (FISH)

*MYC*, *BCL2* or *BCL6* translocation was detected by the FISH, using the break-apart probes of target genes, strictly following manufacturer’s instruction. We tested all samples with the *MYC* probe first, followed by screening for *BCL2* and *BCL6* translocations in samples harboring *MYC* translocation. The probes, namely Vysis LSI MYC (8q24.21) (Cat# 05J91-001), Vysis LSI BCL2 (18q21.33) (Cat# 07J75-001), and Vysis LSI BCL6 (3q27.3) (Cat# 01N23-020) (ASR) dual color break apart rearrangement probes, were purchased from Abbott Laboratories (Chicago, IL, USA). As previously described [18], *MYC*, *BCL2* and *BCL6* were defined as rearranged when they exhibited the appearance of disjointed individual green signal and red signal; For every sample, the probe signals for monolayers of ≥200 DLBCL cell nuclei were counted under fluorescence microscope at ×100 magnification, and probe signals exceeding 20% threshold in the number of nuclei was considered genetic alterations.

### 2.3. Immunohistochemical (IHC) Staining

IHC staining was performed to evaluate the expression level of MYC, BCL2 or BCL6 proteins in our DLBCL samples, using 3 µm formalin-fixed paraffin-embedded (FFPE) tissue sections which were subjected to 3 min. Ethylenediaminetetraacetic acid (EDTA) buffer-based heat-induced antigen retrieval. The samples were probed with monoclonal primary antibodies against CD10 (clone 56C6; Cat# MA5-14050, 1:5, ThermoFisher Scientific, Carlsbad, CA, USA.), IRF4/MUM1 (clone EP190; Cat# BSB-6958, 1:80, BioSB, Santa Barbara, CA, USA), BCL2 (RTU clone SP66; Cat# 790-4604, Roche Tissue Diagnostics, Oro Valley, AZ, USA), MYC (clone EP121; Cat# BSB-6581, 1:30, BioSB), BCL6 (RTU clone GI191E/A8; Cat# 760-4241, Roche Tissue Diagnostics), and Ki67 (clone SP6; Cat# MA5-14520, 1:200, ThermoFisher Scientific). All staining was performed using the Ventana benchmark ULTRA IHC staining module (Ventana, Tucson, AZ, USA). As previously described [19], for COO determination based on Hans’ algorithm, positivity cut-off for CD10, IRF4/MUM1, and BCL6 expression was ≥30% stained cells, while the expression positivity cut-off for BCL2 or MYC was ≥50% or ≥40% of stained cells, respectively.

### 2.4. Statistical Analysis

Correlative analysis and determination of nonrandom association between gene rearrangement or expression and patients’ clinicopathological features were performed by chi-squared (χ^2^) or Fisher exact tests. If the test statistic exceeds the critical value of χ^2^, the null hypothesis (*H*_0_ = there is no difference between the distributions) can be rejected, and the alternative hypothesis (*H*_1_ = there is a difference between the distributions) can be accepted with the selected level of confidence; in order words we can say the variables are dependent/related. Thus, Prob > χ^2^ implies the probability that the test statistic exceeds the critical value of χ^2^ and allows us make the judgement that the null hypothesis can be rejected and the alternative hypothesis accepted, establishing the association between the variables, i.e., the translocation status (*BCL2* DHL, *BCL6* DHL, THL) is related to or associated with survival status. It is important then, that the Prob > χ^2^ and χ^2^ be interpreted in the context of each other, not as standalone values. The overall survival (OS), defined as the period from diagnosis to day of death or last follow-up and progression-free survival (PFS), defined as the period from primary treatment to day of disease worsening and/or recurrence, were estimated using Kaplan-Meier (KM) plots, and the inter-group comparison of survival differences was evaluated using the log-rank test.
Prognostic relevance was determined using univariate and multivariate Cox regression models. Where applicable, compensation for missing data was achieved using the stochastic regression imputation method. All statistical analyses were performed with IBM SPSS Statistics for Windows, Version 20.0 (IBM Corp., Armonk, NY, USA). A
*p*-value < 0.05 was considered statistically significant.


## 3. Results

### 3.1. Cohort Clinical Characteristics, Gene Rearrangement, and Protein Expression

Our cohort of 282 patients with DLBCL consists of 162 males and 120 females with a median age of 66.5 ± 15.2 years. 47 (16.7%) patients harbored *MYC* rearrangement, while 235 (83.3%) exhibited no *MYC* rearrangement. Of the *MYC*-rearranged cases, ~33%, 50%, and ~17% concurrently bore *BCL2*, *BCL6*, or *BCL2*+*BCL6* rearrangement, respectively (Figure 1). Based on COO, 116 (41.1%) of the total cohort were GCB-type, while 166 (58.9%) were non-GCB-type DLBCL (Table 1). Among the 282 patients, 47 (16.7%) had *MYC* translocation, and of these 47 *MYC*-rearranged cases, concurrent *BCL2* and/or *BCL6* translocation were detected in 24 patients; Half of them (12 patients) had simultaneous *MYC* and *BCL6* translocations, eight harbored *MYC* and *BCL2* rearrangements, while the remaining four patients exhibited a triple hits (TH) genotype with concurrent *MYC*, *BCL2* and *BCL6* translocations (Figure 1, Table 2). Together, 16 (66.7%) of DH/TH-HGBL patients were positive for *BCL6* rearrangement (Figure 1). Among these DH/TH-HGBL patients with *BCL6* rearrangement, only six (37.5%) overexpressed MYC and BCL6 proteins simultaneously as shown by IHC staining, indicating the co-overexpression of MYC and BCL6 proteins may not be a plausible surrogate biomarker for screening *BCL6*-rearranged DH HGBL (Table 2). Moreover as shown in Table 2, nine of sixteen (56.3%) *BCL6*-rearranged DH/TH-HGBL cases, and 58.3% of the *BCL6*-rearranged DH-HGBL cases were non-germinal center B-cell (non-GCB), while six of 12 (50.0%) patients with *BCL6*-rearranged DHL had Ann Arbor stages III/IV disease, suggesting that added screening for DH/TH-DLBCL with *BCL6* rearrangement in patients with the non-GCB phenotype of DLBCL may aid medical decision-making and therapeutic strategy.

### 3.2. The Relationship between MYC, BCL2, and BCL6 Rearrangement, Expression and Clinicopathological Features

#### 3.2.1. MYC, BCL2, BCL6, and COO

Compared to patients with *BCL2* DHL, those with *BCL6* DHL were more associated with MUM1 expression (25.0% vs. 66.7%). Conversely, the *BCL2* DHL group was more likely to express CD10/MME protein in comparison with the *BCL6* DHL group (87.5% vs. 50.0%). Consistently, almost three out of every five *BCL6* DHL cases (58.3%) exhibited the non-GCB phenotype, while the *BCL2* DHL group (87.5%) was mostly GCB type. In parallel analysis, following numeralization of *BCL2* DHL, *BCL6* DHL and THL as 1, 2 and 3, respectively, as well as 0 and 1 for non-GCB and GCB phenotypes, respectively, results of our paired *t*-test analysis showed a t-ratio of −6.53 and a correlation index of −1.70, indicating a reversed effect directionality; thus by inference patients with *BCL2* DHL were GCB type, while our *BCL6* DHL and THL were mostly non-GCB type (Figure 2).

#### 3.2.2. MYC, BCL2, BCL6 Rearrangement and Expression

Table 1 and Table 2 summarize the relationships between MYC, BCL2 or BCL6 gene rearrangement and protein expression. Patients with *MYC* rearrangement alone were more likely to express MYC protein (73.9%) than their counterparts with normal *MYC* gene (48.9%); thus, compared with the patients with unaltered *MYC* gene, or those with concomitant *MYC* and *BCL6* rearrangement (41.7%), those harboring concurrent *MYC* and *BCL2* rearrangements more frequently expressed MYC protein (75.0%). More so, patients with rearranged *BCL2* almost always expressed BCL2 protein (87.5%) compared with those with *MYC*-only rearrangement (56.5%) or rearranged *MYC* and *BCL6* but normal *BCL2* genes (58.3%). Interestingly, we observed that all patients with THL, 75.0% of patients with *BCL6* DHL, and 62.5% of *BCL2* DHL cases were BCL6 expressors. From comparative analysis using the paired *t*-test, we found the mean difference between patients with DHL/THL and DEL/TEL was 0.538 ± 0.144 (95%CI: 0.225–0.852) with a t-ratio of 3.742 which is statistically significantly different from 0 at the 95% CI (Prob > *t* = 0.003; Prob > *t* = 0.001; Prob < *t* = 0.999), and correlation of 0 (Figure 3A), indicating that double or triple expressors (DEL/TEL) do not necessarily harbor double or triple hits (DHL/THL).

#### 3.2.3. MYC, BCL2, BCL6, Sex, Age, and Survival

We also observed that the male gender was more predisposed to DHL (*BCL2* DHL: 87.5%, *BCL6* DHL: 58.3%), gender association was equivocal for patients with THL (Male: 50% vs. Female: 50%). Patients with *BCL2* DHL were found to be older (median age: 77 ± 10.8 years) than their *BCL6* DHL (median age: 64.5 ± 11.9 years) or THL (median age: 58 ± 12.4 years) counterparts. Howbeit contextually confounding, we found that with a median survival of 177 ± 122.2 days, patients with *BCL2* DHL exhibited worse overall survival compared to the *BCL6* DHL (598.5 ± 853.7 days) or THL (1381 ± 598.3 days) group.

### 3.3. Gene Rearrangement, Protein Expression, and Proliferation Index

Consistent with the suggestion that the median proliferation index of DHL approaches 90% [11], as indicated in Table 1 and Table 2, we found that added to a median Ki67 index of 90 ± 6.41%, All patients with *BCL6* DHL had a Ki67-based proliferation index ≥ 70%, compared with only 50.0% of the *BCL2* DHL group, 87.0% from the *MYC* only-rearranged group or 76.7% of patients without multiple hits. Interestingly, results of our paired *t*-test showed a correlation of 72.3% between *BCL6* rearrangement and DHL/THL status, with mean difference of −0.15 ± 0.05 and t-ratio of −3.04 (Figure 3B). This data does suggest that while DLBCL cells are characteristically highly proliferative, a role for BCL6 as ‘driver’ or ‘enhancer’ of this highly proliferative phenotype especially in DHL cases cannot be disregarded. High proliferation index in the *MYC* only-rearranged and *BCL6* DHL cases were mostly associated with low or equivocal CD10 expression and co-immunopositivity of MUM1, MYC, BCL2 and BCL6 protein expression, and these patients were more likely to be non-GCB type DLBCL.

### 3.4. Prognostic Relevance of MYC, BCL2, or BCL6 Rearrangement and Expression

The median follow-up duration for our whole cohort (*n* = 282) was 465 months, ranging from 0 to 3835 days, while it was 484 ± 753.3 days for patients with DHL/THL. Regardless of rearrangement status, there were 108 (38.3%) recorded deaths; 22 (7.8%) of these were secondary to non-DLBCL-related causes, 70 patients (24.8%) were disease-specific mortality cases while the remaining 16 patients were without recorded cause of death. 8.5% (24/282) of the whole DLBCL cohort were patients with documented recurrent disease. Survival analyses showed that patients with DHL/THL had shorter overall survival (OS) than their non-DHL/THL counterparts (χ^2^ = 3.50, Prob > χ^2^ = 0.06) (Figure 4A). Upon stratification by hits, by the end of year 5, all patients with THL, *BCL2* DHL and all but one *BCL6* DHL cases had expired or were lost to follow-up (χ^2^ = 8.56, Prob > χ^2^ = 0.01) (Figure 4B). When age-adjusted, for patients with DHL/THL, age <60 years (*n* = 90) conferred a survival advantage of 500 days (~17 months) compared to those aged ≥ 60 years (*n* = 192) (Figure 4C,D). Similarly, progression-free survival (PFS) was longer for the non-DHL/THL group compared with the DHL/THL group (χ^2^ = 2.01, Prob > χ^2^ = 0.16) (Figure 4E,F); More so, while the six patients with *BCL2* DHL were lost to follow-up by day 800, their remaining THL and *BCL6* DHL peers exhibited very poor PFS, regardless of age strata (Figure 4F, Appendix A). Moreover, double and triple expressors (DEL/TEL) were less likely to be alive by day 2600 (86.7 month), compared with their non-DEL/TEL counterparts who enjoyed survival advantage of~900 days (~30 months; χ^2^ = 1.9, Prob > χ^2^ = 0.16) (Appendix A) and longer PFS (χ^2^ = 1.15, Prob > χ^2^ = 0.28) (Appendix A). We also observed 19 cases of extranodal involvement; 68.4% and 52.6% of which were *BCL6*-, and *BCL2*- rearranged, respectively (Appendix A). More than twice as much patients without *BCL6* translocation (*n* = 6) harbored *BCL6* rearrangement (*n* = 13) and exhibited extranodal involvement. Moreover, patients with *BCL6* rearrangement were 5.5-fold more likely associated with extranodal involvement compared with their *BCL2*-rearranged peers ((*BCL6*: HR(95%CI) = 3.8 (0.64–22.2) vs. *BCL2*: HR(95% CI) = 0.6847 (0.2265 to 2.0701)), and our variable-outcome analysis indicate that while *BCL2* rearrangement exhibited no statistically relevant relationship with extranodal involvement (χ^2^ = 0.50, *p* = 0.48), *BCL6* hits showed a significantly strong relationship with extranodal involvement (χ^2^ = 5.79, *p* = 0.02) (Appendix A).


## 4. Discussion

DLBCL is a particularly aggressive disease entity which is broadly characterized by recurrent gene aberration, including the relatively infrequent presence of *MYC* (8q24), *BCL2* (18q21), and/or *BCL6* (3q27) gene translocations/rearrangements [11,12,13,14,15,16,17]. Despite our increased understanding of DHL/THL biology and advances made in the therapeutic management of these therapy-evasive HGBL with *MYC* and *BCL2* and/or *BCL6* rearrangements, the nosological characterization of DHL/THL remains incomplete, necessitating transnational or multicenter large cohort studies to unravel relevant disease-specific genetic and clinicopathological features, as well as risk factors for appropriate characterization of risk-adapted therapeutic strategies [11,12,13,14,15,16,20]. More so, while acknowledging the poor prognostic peculiarity of DHL/THL and the benefits associated with inclusion of MYC, BCL2, and BCL6 gene translocation and protein expression in routine clinical work-up of patients with suspected DLBCL, little is known about the differential ethno-specific or geo-regional signature of DHL/THL. In fact, while the clinicopathological characteristics and probable therapeutic options of *BCL2*-rearranged DHL are well documented, only few studies have focused on *BCL6*-rearranged DHL and its clinical features.

Against the background of suggested geographic variation in the molecular pathogenesis of lymphoma [21], in the present study, exploring for probable Taiwan-specific clinicopathological and cytogenetic characteristics of DLBCL, we observed that 8.5% (24/282) of our DLBCL cohort were HGBL with re-arranged *MYC* and *BCL2* and/or *BCL6* genes, which is consistent with earlier FISH-based studies indicating that 7–10% patients with DLBCL harbor *MYC*, *BLC2* and/or *BCL6* rearrangement [11,13]; however compared with the estimated percentage in most published works [11,13], we report a higher prevalence of *MYC* translocation prevalence in our cohort (~10% vs. 16.5%).

Similar to the documented predominance of the GCB subtype in global DHL/THL cases (more specifically, from western countries) [13,22,23], our Taiwanese DHL/THL cohort were predominantly GCB phenotype. Interestingly and of clinical relevance, the present study reports the predominance of *BCL6*-rearranged DHL and a 3-to-2 *BCL6*-to-*BCL2*-rearranged DHL ratio, in contrast to the broadly reported *BCL2* predominant DHL and the global *BCL6*-to-*BCL2* ratio ranging from 1:1 to 1:8 [11]. We posit that this enhanced *BCL6*:*BCL2* ratio and predominance of *BCL6* rearrangement among patients with DHL is not only suggestive of a selective complementary role for *MYC* and *BCL6* among Taiwanese patients, but may be associated with the poor therapy response and aggressive clinical course of our patients with DHL. This is even more so considering that these *BCL6*-rearranged DHL were immunophenotypically unlike their *BCL2* DHL counterpart, being principally non-GCB (immunoblastic, ABC type) in our cohort. This is consistent with reports indicating that patients with *BCL6* DHL are less likely immunophenotypically GCB, clinically aggressive, characterized by poor OS, and that the “ABC DLBCL is associated with substantially worse outcomes when treated with standard chemoimmunotherapy”, unlike their GCB peers with better therapeutic outcomes [14,24]. On the contrary, a number of studies (originating mostly from western countries) suggest that DLBCL “patients with either *MYC/BCL6* rearrangements or MYC/BCL6 co-expression did not always have poorer prognosis”, including report that *BCL6*-rearranged DHL cases are phenotypically GCB, and exhibit markedly better survival rates compared to their *BCL2* DHL counterparts, who were found to be largely ABC in phenotype [25]. Interestingly, Li et al., using a similar Caucasian cohort as the study alluded above, found no apparent difference in the clinicopathological characteristics of patients with *BCL6* DHL and those with *BCL2* DHL, except that the *BCL6* DHL group were less likely to be GCB in immunophenotype and had poor overall survival [14].

In addition, we observed that all our patients with *BCL6* DHL had a Ki67 proliferation index ≥ 70, compared with half of the *BCL2* DHL group, suggesting a critical role for *BCL6* in the highly proliferative phenotype of our DHL cases. This finding is consistent with reports indicating that the constitutively pro-oncogenic BTB/POZ domain-containing BCL6, a transcriptional repressor of apoptosis, inflammation, and cell cycle control-effector genes [11], is required for B-cell proliferation, represses replication checkpoints, enhances tolerance to DNA damage [26,27]. Moreover, the high proliferation index of our *BCL6* DHL cases is associated with low/equivocal CD10 expression and co-immunopositivity of MUM1, MYC, BCL2 and BCL6 protein expression, and these patients were more likely to be non-GCB type DLBCL. Concordant with accruing evidence, our finding highlights the essential role of *BCL6* in the survival of DLBCL regardless of COO, and perhaps explains why targeting BCL6 is equally efficacious for suppression of both the very aggressive non-GCB and relatively less aggressive GCB type DHL [28].

Our study also demonstrated that Taiwanese patients with *BCL6* DHL are not necessarily MYC protein expressors, but are more often than not BCL6 expressors. Against the background that BCL2 and MYC are mostly co-expressed, our finding is congruent with Pillai RK et al.’s observation that unlike *BCL2* DHL cases, *BCL6* DHL are more likely to be CD10- but IRF4/MUM1+ and, more like Burkitt lymphoma, are cytogenetically less complex, and only infrequently express BCL2 [29], and by inference, less often express MYC. This may be associated with reported self-mitigating propensity of BCL6, a modulator of B cell receptor signals, to transcriptionally repress oncogenes such as MYC, BCL2, cyclin D1 (CCND1), and B lymphoma Mo-MLV Insertion Region 1 homolog (BMI1) [30,31]. It is thus evident and mechanistically relevant that *BCL6* rearrangements are not associated with disruption of *BCL6* coding sequence, but does replace the *BCL6* promoter sites with other promoters; this allows BCL6 transformation-associated malignant transformation of B cells through dysregulated expression of normal BCL6 protein [30]. Concordant with the hypothesis of this present study, it is scientifically plausible that “because BCL6 represses beta-interferon gene positive-regulatory domain 1 binding factor (*BLIMP1*), which in turn represses *MYC*, *BCL6* translocations might be considered functionally equivalent to *MYC* translocation” [31]. Consistently, while our THL are equivocal for age or distribution, Ann-Arbor staging, Ki-67, CD10, MUM1, and BCL2 proteins expression, they are predominantly MYC expressors (75%) and wholly BCL6 expressors (100%). Thus, our study results portray BCL6 as a principal determinant of HGBL-DH/TH in Taiwan. The predominance of *BCL6* rearrangement and associated aberrant expression of BCL6, BCL2, MUM1, and MYC proteins in Taiwanese patients with DHL in contrast to *BCL6* DHL rarity in the West informs medical decision making, and does suggest that BCL6 is a promising therapeutic target [32,33]. There are reports demonstrating the druggability of BCL6 and the strong antiproliferative effect of its degradation [34], as well as evidence that the conditional deletion of BCL6 in DLBCL tumors in vivo induced significant inhibition of tumor growth with initial tumor stasis and subsequent attenuated tumor growth kinetics [35]. This is of contextual relevance and lends some credence to the findings of our present study which makes a case for the routine screening for BCL6 gene rearrangement by FISH and protein expression by IHC in all newly diagnosed DLBCL cases in Taiwan, and recommends the molecular or pharmacological targeting of BCL6 as an efficacious therapeutic strategy in managing patients with aggressive DHL regardless of COO in Taiwan.

It is worth mentioning that though initially confounding, the seemingly worse survival among *BCL2* DHL in the present study is attributable to the larger number of their constituent patients in advanced stage with 62.5% stage III/IV and 37.5% stage I/II, compared with the *BCL6* DHL group with equiproportional cases in early and late stages. More so, these patients with *BCL2* DHL were more advanced in age with median age of 77 ± 10.8 years. Aside these risk factor, consistent with the well documented role of BCL6 in the initiation, therapy evasion, and progression of hematological malignancies, including DLBCL [11,26,27,28,29,30,31,32,33,34,35], it is conceivable that this worse prognosis is associated with the high expression of BCL6 protein in most of these *BCL2* DHL cases (62.5% BCL6+ vs. 37.5% BCL6−). Thus, we posit that this observed poor prognosis may be attributable to aberrant BCL6 expression and not because of the BCL2 translocation *per se*, in the *BCL2* DHL cases.

## 5. Conclusions

In conclusion, contrary to contemporary data from western countries, the majority of patients with multiple hits-DLBCL in Taiwan harbor *BCL6* rearrangement. While IHC-based co-overexpression of MYC and BCL2 proteins is a good surrogate for *BCL2*-rearranged DHL, same may not be said for *BCL6*-rearranged DH/TH-DLBCL in Taiwan. More than half of our *BCL6*-rearranged DH-DLBCL cases were non-GCB phenotype, indicating, at least in part, that the preferential screening for DH with *BCL6* rearrangement may be a clinically-informative modality for patients with the non-GCB phenotype DLBCL in Taiwan. Consistent with present findings, we recommend mandatory screening for *BCL6*-rearranged DH DLBCL in Taiwan.

## Figures and Tables

**Figure 1 cancers-13-01620-f001:**
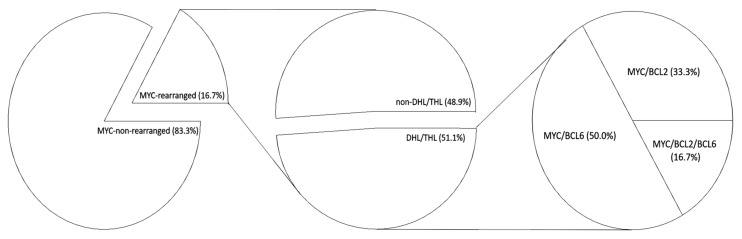
Pie-charts showing our DLBCL cohort (*n* = 282) distribution based on the presence or absence of MYC-rearrangement (*left*), and multiple hit status (*middle*), as well as the stratification of the multiple hit group based on concurrent BCL2 and/or BCL6 rearrangements (*right*). The ratio of rearranged BCL2:BCL6 was 2:3. DHL, double hit lymphoma; THL, triple hit lymphoma.

**Figure 2 cancers-13-01620-f002:**
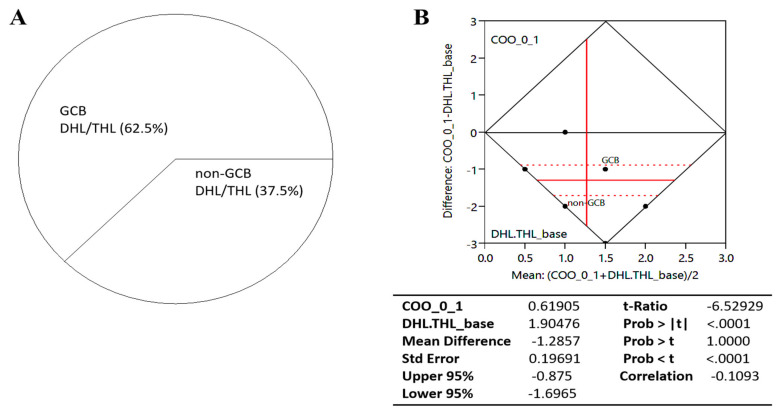
(**A**) Pie-chart showing the COO-based constitution of our high-grade B-cell lymphoma (HGBL) with *MYC* and *BCL2* and/or *BCL6* rearrangements (*n* = 24). (**B**) Chart showing the difference between the means of our high-grade B-cell lymphoma (HGBL) with *MYC* and *BCL2* and/or *BCL6* rearrangements (DHL/THL) and their COO. DHL, double hit lymphoma; THL, triple hit lymphoma; DEL, double expressor lymphoma; TEL, triple expressor lymphoma; N, sample size; DF, degree of freedom; GCB, germinal center B-cell type; COO, cell-of-origin.

**Figure 3 cancers-13-01620-f003:**
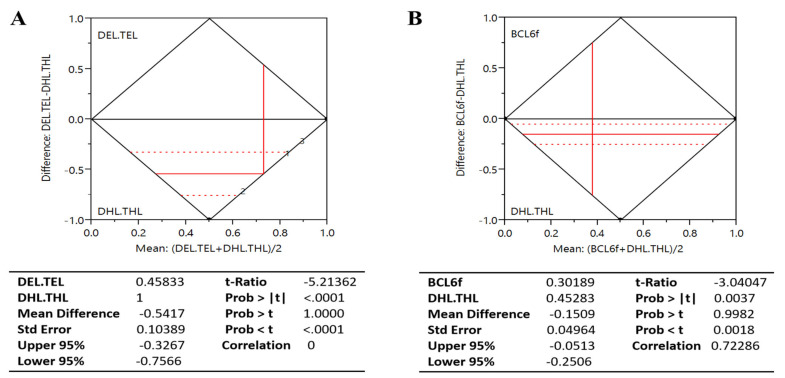
Patients with DHL or THL mostly harbor BCL6 rearrangement, are mostly BCL6 expressors, but are not necessarily double or triple expressors. (**A**) Chart showing the difference between the means of our high-grade B-cell lymphoma (HGBL) with *MYC* and *BCL2* and/or *BCL6* rearrangements (DHL/THL) and those with MYC and BCL2 and/or BCL6 protein expression (DEL/TEL). (**B**) Chart showing the difference between the means of our high-grade B-cell lymphoma (HGBL) with *MYC* and *BCL2* and/or *BCL6* rearrangements (DHL/THL) and those with BCL6-rearrangement (BCL6f). DHL, double hit lymphoma; THL, triple hit lymphoma; DEL, double expressor lymphoma; TEL, triple expressor lymphoma; BCL6f, BCL6-rearranged.

**Figure 4 cancers-13-01620-f004:**
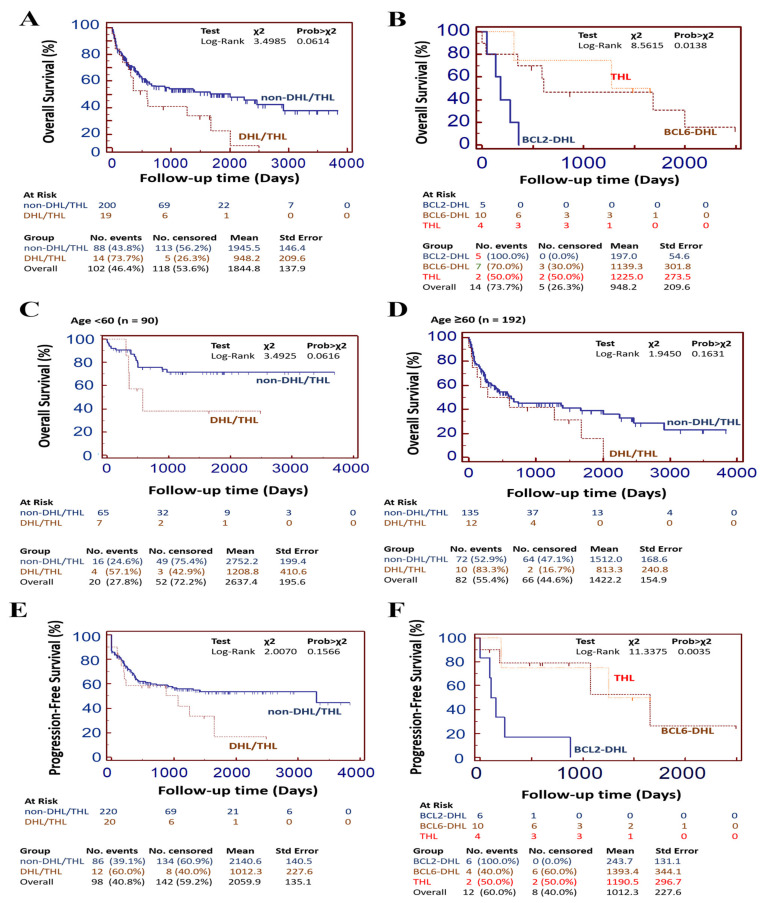
Multiple hits (DHL/THL) confer poor prognosis. Kaplan-Meier curves showing the differential effect of (**A**) DHL/THL or non-DHL/THL, and (**B**) BCL2-DHL, BCL6-DHL or THL on the overall survival of our DLBCL cohort (*n* = 282). Kaplan-Meier curves showing the age-adjusted differential effect of DHL/THL or non-DHL/THL on the overall survival of our DLBCL cohort (*n* = 282) in patients (**C**) younger than, or (**D**) older than 60 years. Kaplan-Meier curves showing the differential effect of (**E**) DHL/THL or non-DHL/THL, and (**F**) BCL2-DHL, BCL6-DHL or THL on the Progression-free survival of our DLBCL cohort (*n* = 282). DHL, double hit lymphoma; THL, triple hit lymphoma; *X*^2^, chi-square.

**Table 1 cancers-13-01620-t001:** Clinicopathological characteristics of our DLBCL cohort (*n* = 282) based on the presence or absence of gene rearrangement.

		Gene Rearrangement/Translocation			
		Non-DHL/THL (Non-MYC + (BCL2 ± BCL6))	DHL/THL (MYC + (BCL2 ± BCL6))	MYC Only	*p*-Value
		*N* = 258 (91.5%)	*N* = 24 (8.5%)	*N* = 23 (8.2%)	
Median Age ± SD (years)		66 ± 15.42	67.5 ± 12.43	64 ± 16.93	
Age (%)					0.651
	<60	83 (32.2%)	7 (29.2%)	10 (43.5%)	
	≥60	175 (67.8%)	17 (70.8%)	13 (56.5%)	
Ann-Arbor stage (%)					0.511
	Early (I/II)	106 (41.1%)	9 (37.5%)	11 (47.8%)	
	Advanced (III/IV)	152 (58.9%)	15 (62.5%)	12 (52.2%)	
Sex (%)					0.345
	Male	146 (56.6%)	16 (66.7%)	14 (60.9%)	
	Female	112 (43.4%)	8 (33.3%)	9 (39.1%)	
Ki67 expression					0.789
	<70%	60 (23.3%)	6 (25.0%)	3 (13.0%)	
	≥70%	198 (76.7%)	18 (75.0%)	20 (87.0%)	
CD10 expression					0.003
	Positive	79 (30.6%)	15 (62.5%)	10 (43.5%)	
	Negative	179 (69.4%)	9 (37.5%)	13 (56.5%)	
MUM1 expression					0.003
	Positive	202 (78.3%)	13 (54.2%)	14 (60.9%)	
	Negative	56 (21.7%)	11 (45.8%)	9 (39.1%)	
MYC expression					0.658
	Positive	131 (50.8%)	14 (58.3%)	17 (73.9%)	
	Negative	127 (49.2%)	10 (41.7%)	6 (26.1%)	
BCL2 expression					0.026
	Positive	215 (83.3%)	16 (66.7%)	13 (56.5%)	
	Negative	43 (16.6%)	8 (33.3%)	10 (43.5%)	
BCL6 expression					0.051
	Positive	182 (70.5%)	18 (75.0%)	13 (56.5%)	
	Negative	76 (29.5%)	6 (25.0%)	10 (43.5%)	
IPI score					0.674
	0–1	81 (31.4%)	5 (20.8%)	10 (43.5%)	
	2	57 (22.1%)	9 (37.5%)	3 (13.0%)	
	3	48 (18.6%)	6 (25.0%)	6 (26.1%)	
	4	43 (16.7%)	4 (16.7%)	1 (4.3%)	
	5	29 (11.2%)	0 (00.0%)	3 (13.0%)	
Cell-of-Origin (COO)					0.007
	GCB	101 (39.1%)	15 (62.5%)	12 (52.2%)	
	Non-GCB	157 (60.9%)	9 (37.5%)	11 (47.8%)	
Follow-up time (days, median ± SD)		462.5 ± 879.19	484 ± 753.31	228 ± 789.45	0.883

**Table 2 cancers-13-01620-t002:** *MYC*, *BCL2* and *BCL6* rearrangement-stratified clinicopathological characteristics of our DLBCL cohort (*n* = 282) based on the presence or absence of gene rearrangement.

		MYC + (BCL2 ± BCL6) Gene Rearrangement/Translocation			
		MYC + BCL2 + BCL6	MYC + BCL2	MYC + BCL6	*p*-Value
		*N* = 4 (16.7%)	*N* = 8 (33.3%)	*N* = 12 (50.0%)	
Median Age ± SD (Years)		58 ± 12.37	77 ± 10.77	64.5 ± 11.93	
Age (%)		0.365			
	<60	2 (50.0%)	1 (12.5%)	4 (33.3%)	
	≥60	2 (50.0%)	7 (87.5%)	8 (66.7%)	
Ann-Arbor Stage (%)		0.856			
	Early (I/II)	2 (50.0%)	3 (37.5%)	6 (50.0%)	
	Advanced (III/IV)	2 (50.0%)	5 (62.5%)	6 (50.0%)	
Sex (%)		0.296			
	Male	2 (50.0%)	7 (87.5%)	7 (58.3%)	
	Female	2 (50.0%)	1 (12.5%)	5 (41.7%)	
Ki67 Expression		0.069			
	<70%	2 (50.0%)	4 (50.0%)	0	
	≥70%	2 (50.0%)	4 (50.0%)	12 (100%)	
CD10 Expression		0.271			
	Positive	2 (50.0%)	7 (87.5%)	6 (50.0%)	
	Negative	2 (50.0%)	1 (12.5%)	6 (50.0%)	
MUM1 Expression		0.275			
	Positive	2 (50.0%)	2 (25.0%)	8 (66.7%)	
	Negative	2 (50.0%)	6 (75.0%)	4 (33.3%)	
MYC Expression		0.322			
	Positive	3 (75.0%)	6 (75.0%)	5 (41.7%)	
	Negative	1 (25.0%)	2 (25.0%)	7 (58.3%)	
BCL2 Expression		0.376			
	Positive	2 (50.0%)	7 (87.5%)	7 (58.3%)	
	Negative	2 (50.0%)	1 (12.5%)	5 (41.7%)	
BCL6 Expression		0.295			
	Positive	4 (100.0%)	5 (62.5%)	9 (75.0%)	
	Negative	0	3 (37.5%)	3 (25.0%)	
IPI Score		0.132			
	0–1	0	2 (25.0%)	3 (25.0%)	
	2	2 (50.0%)	0	6 (50.0%)	
	3	2 (50.0%)	3 (37.5%)	1 (8.3%)	
	4	0	3 (37.5%)	1 (8.3%)	
	5	0	0	1 (8.3%)	
Cell-of-Origin (COO)		0.351			
	GCB	3 (75.0%)	7 (87.5%)	5 (41.7%)	
	Non-GCB	1 (25.0%)	1 (12.5%)	7 (58.3%)	
Follow-up time (days, median ± SD)		1381 ± 598.33	177 ± 122.18	598.5 ± 853.70	0.088

## Data Availability

The data presented in this study are available on request from the corresponding author. The data are not publicly available due to patient confidentiality and institutional data publication policy.

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
