# Peer review of "High-Grade B-Cell Lymphoma (HGBL) with MYC and BCL2 and/or BCL6 Rearrangements Is Predominantly BCL6-Rearranged and BCL6-Expressing in Taiwan"

_cancers, 2021, doi:10.3390/cancers13071620_

Round 1

Reviewer 1 Report

This is an interesting study which can add value to this research area. Comments:

  1. The authors should discuss the potential mechanisms of less frequency of MYC expression in MYC/BCL6 double-hit cases. One plausible explanation is that BCL6 suppresses MYC expression at transcription level. There are literatures on this.
  2. The molecular classification of DLBCL evolves rapidly. GCB and ABC classification showed its value in early studies as the authors mentioned in Introduction, but some recent studies showed less to no value in terms of prognosis. The authors should cite literatures and discuss. In addition, recent studies have classified DLBCL into 4 prominent genetic subtypes (NEJM, April 2018). The authors should discuss this too.
  3. P values are needed in Table 1 and 2
  4. The authors mentioned “most BCL6 DHL cass (58.3%)….”. 58.3% is about half and not most to me. The authors may consider to change the word most.
  5. There is controversy in terms of the prognosis of BCL6/MYC double-hit patients in the literatures. Some showed a worse survival whereas other did not. The author should cite the paper by Young et al in Oncotarget. 2016; 7:2401-2416. For the current study, from Figure 4, it seems to me that BCL6-DHL has a better survival than BCL2-DHL, which not surprised me. The author should elaborate clearly and extensively on the impact of BCL6-DHL on survival. Any result (BCL6-DHL carries worse survival or not) is interesting and valuable.
  6. Prognosis should be mentioned in abstract.
  7. Language should be polished a bit.

Reviewer 2 Report

Minors comments : 

  • introduction should be revised : please introduce more clearly what are the objectives of the study
  • how do you explain that BCL6 expressors are found among BCL2 double hits patients?
  • last paragraph of discussion (starting line 365) should be revised. Conclusion line 374-376 relies on assumptions based on little data. This should therefore be reformulated.

Round 2

Reviewer 1 Report

  1. I am still confused by the statistical analysis performed in Fig 4.  Fro some reason, the authors use probability > chi square to illustrate the survival difference.  Can the authors please clarify in the methods how to define significantly difference?  For example in Fig 4A, P is 0.06, is this significant? And in Fig 4B, p is 0.01, is this not significant?
  2. How the authors explain the survival difference between BCL2-DHL and THL/BCL6-DHL?
  3. In page 12, "BCL6 translocations might be considered functionally equivalent to MYC translocation". I do not think so.

Round 3

Reviewer 1 Report

None